Accepted at the ICLR 2024 Workshop on AI4Differential Equations In Science

# PHYSICS-INFORMED NEURAL NETWORKS FOR SAMPLING

**Jingtong Sun,**[*] **Julius Berner,**[*] **& Anima Anandkumar**
California Institute of Technology
{jeffsun, jberner, anima}@caltech.edu

**Kamyar Azizzadenesheli**
NVIDIA
kamyara@nvidia.com

## ABSTRACT

We present a framework to sample from high-dimensional unnormalized densities using physics-informed neural networks (PINNs). For various computational science tasks, it is essential to draw samples from a target distribution where the density is known up to a normalizing constant. Without access to any training samples, existing methods based on normalizing flows and diffusion models rely on the simulation of (stochastic) differential equations for training and suffer from mode collapse. Our approach circumvents these issues by solving the underlying continuity and Fokker-Planck equations using PINNs. Motivated by optimal transport and Schrödinger bridges, we further incorporate regularizers based on Hamilton-Jacobi-Bellman equations. Through evaluations on several benchmarks, we demonstrate that our approach can mitigate mode collapse and significantly outperform various baselines.

## 1 SAMPLING WITH STOCHASTIC PROCESSES

We are given a strictly positive, sufficiently smooth function $q_{\text{target}} \colon \mathbb{R}^d \to (0, \infty)$ and our goal is to sample from the density

$$p_{\text{target}} = \frac{q_{\text{target}}}{Z}, \quad \text{where} \quad Z \coloneqq \int q_{\text{target}}(x) \, \mathrm{d}x. \tag{1}$$

To achieve this goal, we want to construct stochastic processes $(X_t)_{t \in [0,T]}$ controlled by neural networks such that

$$X_T \sim p_{\text{target}}. \tag{2}$$

Typically, the process $X$ is assumed to start at a tractable prior density $X_0 \sim p_{\text{prior}}$, for instance a standard Gaussian $p_{\text{prior}} = \mathcal{N}(0, \mathrm{I})$. Moreover, we assume that $X$ is governed by a *stochastic*[1] *differential equation* (SDE)

$$\mathrm{d}X_t = \mu(X_t, t) \, \mathrm{d}t + \sigma(t) \, \mathrm{d}B_t, \quad X_0 \sim p_{\text{prior}}, \tag{3}$$

where $\sigma \colon [0, T] \to \mathbb{R}^{d \times d}$ is a given diffusion coefficient and the drift $\mu \colon \mathbb{R}^d \times [0, T] \to \mathbb{R}^d$ is parametrized by neural networks. We also might have additional constraints on the trajectories of $X$.

Recently, several such methods based on SDEs (nonzero $\sigma$) as well as ODEs ($\sigma \equiv 0$) have been established. The former are, for instance, based on *Schrödinger (half-)bridges*, *diffusion models*, or annealed flows (Vargas & Nüsken, 2023; Zhang & Chen, 2022; Berner et al., 2022; Richter et al., 2023; Zhang et al., 2023; Vargas et al., 2023). The latter leverage continuous-time *normalizing flows* and combinations with *Monte Carlo* (MCMC) methods (Wu et al., 2020; Midgley et al., 2022; Matthews et al., 2022; Arbel et al., 2021), see Appendix A for further related work. However, all the previously mentioned methods rely on simulating (parts of) the process $X$ for training. This requires time-discretizations and typically results in unstable and slow convergence.

---

[*]Equal contribution.

[1]We will also consider the case where the diffusion coefficient $\sigma$ is zero. The evolution in (3) then corresponds to an *ordinary differential equation* (ODE), however $X$ is still a *stochastic* process due to its random initial condition given by $p_{\text{prior}}$.

In this work, we identify the partial differential equations (PDEs) governing the underlying dynamics of many of these methods. Leveraging neural PDE solver, such as *physics-informed neural networks* (PINNs) (Raissi et al., 2017; Sirignano & Spiliopoulos, 2018), we show that this leads to simulation- and discretization-free objectives. We compare various objectives and demonstrate strong performance on challenging, high-dimensional distributions.

## 2 LEARNING THE EVOLUTION

Assuming that $X_t$ has a sufficiently smooth density $p_X(\cdot, t)$, we first note that the density satisfies the *Fokker-Planck equation*

$$\partial_t p_X = -\operatorname{div}(p_X \mu) + \tfrac{1}{2}\operatorname{Tr}(\sigma\sigma^\top \partial_{xx} p_X), \quad p_X(\cdot, 0) = p_{\text{prior}}. \tag{4}$$

For numerical stability, it is convenient to work with the log-density $v := \log p_X$ satisfying the equation

$$\partial_t v = P_{\text{FP}}^{\sigma,\mu}(v), \quad \text{with} \quad P_{\text{FP}}^{\sigma,\mu}(v) := -\operatorname{div}(\mu) - \nabla v \cdot \mu + \tfrac{1}{2}\|\sigma^\top \nabla v\|^2 + \tfrac{1}{2}\operatorname{Tr}(\sigma\sigma^\top \partial_{xx} v). \tag{5}$$

This nonlinear PDE can be viewed as a *Hamilton-Jacobi-Bellman* (HJB) equation and its derivation follows from the *Hopf-Cole transform*, see also Evans (2010). Motivated by Máté & Fleuret (2023), we want to use the unnormalized density $q_{\text{target}}$ in our parametrization of the log-density $v_{w,z}$, i.e.,

$$v_{w,z}(\cdot, t) = \tfrac{t}{T}\log\tfrac{q_{\text{target}}}{z(t)} + \left(1 - \tfrac{t}{T}\right)\log p_{\text{prior}} + \tfrac{t}{T}\left(1 - \tfrac{t}{T}\right)w(\cdot, t), \tag{6}$$

where $w$ and $z$ are neural networks approximating the log-density and normalizing constant.

If $v_{w,z}$ satisfies (5), i.e., $\partial_t v_{w,z} = P_{\text{FP}}^{\sigma,\mu}(v_{w,z})$, conservation of mass implies that $z(T) = Z$ and thus the terminal condition $v_{w,z}(\cdot, T) = \log p_{\text{target}}$ is satisfied. Penalizing the square of the PDE residual for $v_{w,z}$ in (5), this yields the loss

$$\mathcal{L}_{\text{FP}}(w, z, \mu) := \mathbb{E}\left[\left(\partial_t v_{w,z}(\xi, \tau) - P_{\text{FP}}^{\sigma,\mu}(v_{w,z})(\xi, \tau)\right)^2\right], \tag{7}$$

where $(\xi, \tau)$ is a suitable random variable distributed on $\mathbb{R}^d \times [0, T]$. As commonly done for PINNs, we chose $\tau \sim \text{Unif}([0, T])$ and $\xi \sim \text{Unif}(K)$ for a sufficiently large compact set $K \subset \mathbb{R}^d$ (that can potentially depend on $\tau$). We refer to Appendix E.1 for other approaches.

*Remark* 2.1 (Deterministic vs. stochastic evolutions). Setting the diffusion coefficient $\sigma$ to zero in (3) yields an *ordinary differential equation* (ODE) with a *deterministic* evolution. When the drift $\mu$ is controlled by a neural network, this is referred to as *continuous-time normalizing flow* (Papamakarios et al., 2021; Rezende & Mohamed, 2015). The PDE in (4) reduces to the *continuity equation*

$$\partial_t p_X = -\operatorname{div}(p_X \mu), \quad p_X(\cdot, 0) = p_{\text{prior}}, \tag{8}$$

and the log-density $v$ satisfies that

$$\partial_t v = -\operatorname{div}(\mu) - \nabla v \cdot \mu. \tag{9}$$

This is computationally cheaper than the stochastic case $\sigma \neq 0$, where the computation of second-order derivatives of $v$ is required for the term $\tfrac{1}{2}\operatorname{Tr}(\sigma\sigma^\top \partial_{xx} v)$. Motivated by the success of diffusion models (Ho et al., 2020; Kingma et al., 2021; Nichol & Dhariwal, 2021; Vahdat et al., 2021; Song & Ermon, 2020) and their interpretation based on SDEs (Song et al., 2020), we still want to explore whether the additional noise can be beneficial.

### 2.1 CONSTRAINING THE EVOLUTION

There are infinitely many combinations of drifts $\mu$ and log-densities $v_{w,z}$ that define a stochastic evolution from the prior to the target distribution, i.e., minimize the loss in (7). The next paragraphs show how we can adapt our loss to yield a unique solution.

**Annealed flows:** As already observed in Máté & Fleuret (2023), we can just define $w \equiv 0$ to anneal between $p_{\text{prior}}$ and $q_{\text{target}}$, bearing similarity with *Annealed Importance Sampling* (AIS) (Neal, 2001) and *Controlled Monte Carlo Diffusions* (CMCD) (Vargas & Nüsken, 2023). Under mild conditions, we can find both an ODE or an SDE governed by a potential $\mu = \nabla\Phi$ that has the prescribed marginals, see Neklyudov et al. (2022). Accordingly, this yields the loss

$$\mathcal{L}_{\text{anneal}}(z, \mu) := \mathcal{L}_{\text{FP}}(0, z, \mu). \tag{10}$$

**Diffusion models:** Let us now consider a non-zero diffusion coefficient $\sigma$ in (3). Moreover, we consider the drift $\mu = \sigma\sigma^\top \nabla v_{w,z} - f$ for a suitable function $f$ to be specified later. Plugging it into the differential operator in (5), we see that

$$P_{\mathrm{FP}}^{\sigma,\mu}(v_{w,z}) = \mathrm{div}(f) + \nabla v_{w,z} \cdot f - \tfrac{1}{2}\|\sigma^\top \nabla v_{w,z}\|^2 - \tfrac{1}{2}\mathrm{Tr}(\sigma\sigma^\top \partial_{xx} v_{w,z}) = -P_{\mathrm{FP}}^{\sigma,f}(v_{w,z}). \quad (11)$$

If $v_{w,z}$ satisfies $\partial_t v_{w,z} = P_{\mathrm{FP}}^{\sigma,\mu}(v_{w,z})$, we thus have that

$$\partial_t \breve{v}_{w,z} = -P_{\mathrm{FP}}^{\breve{\sigma},\breve{\mu}}(\breve{v}_{w,z}) = P_{\mathrm{FP}}^{\breve{\sigma},\breve{f}}(\breve{v}_{w,z}), \quad (12)$$

where we write $\breve{\sigma}$ for the time-reversal, i.e., $\breve{\sigma}(t) = \sigma(T - t)$. This means that we can identify $\breve{v}_{w,z} = \log p_Y$ as the log-density of the process

$$\mathrm{d}Y_t = \overleftarrow{f}(Y_t, t)\,\mathrm{d}t + \breve{\sigma}(t)\,\mathrm{d}B_t, \quad Y_0 \sim p_{\mathrm{target}}, \quad (13)$$

as also derived in Berner et al. (2022). Thus, a viable strategy is to pick $f$ and $\sigma$ such that $p_Y(\cdot, T) \approx p_{\mathrm{prior}}$, see Song et al. (2020) for suitable choices, and minimize the loss

$$\mathcal{L}_{\mathrm{diff}}(w, z) := \mathcal{L}_{\mathrm{FP}}(w, z, \sigma, \sigma\sigma^\top \nabla v_{w,z} - f) = \mathbb{E}\left[\left(\partial_t v_{w,z}(\xi, \tau) + P_{\mathrm{FP}}^{\sigma,f}(v_{w,z})(\xi, \tau)\right)^2\right]. \quad (14)$$

**Optimal transport and Schrödinger bridges:** We can also seek the drift $\mu$ that additionally minimizes an energy of the form

$$\inf_\mu \; \tfrac{1}{2}\int_0^T \mathbb{E}\left[\|\mu(X_s, s)\|^2\right]\,\mathrm{d}s. \quad (15)$$

For $\sigma = 0$, this is connected to *optimal transport* (OT) problems w.r.t. the Wasserstein metric (Benamou & Brenier, 2000). For nonzero $\sigma$, this corresponds to the dynamic *Schrödinger bridge* (SB) problem (Dai Pra, 1991). In these cases, the optimal solution can be written as $\mu := \nabla\Phi$, where $\Phi$ solves the *Hamilton-Jacobi-Bellman* (HJB) equation

$$\partial_t \Phi = P_{\mathrm{HJB}}^\sigma \Phi, \quad \text{with} \quad P_{\mathrm{HJB}}^\sigma \Phi = -\tfrac{1}{2}\|\nabla\Phi\|^2 - \tfrac{1}{2}\mathrm{Tr}(\sigma\sigma^\top \partial_{xx}\Phi), \quad (16)$$

see Appendix B. We can add such regularization using the loss

$$\mathcal{L}_{\mathrm{HJB}}(\Phi) := \mathbb{E}\left[(\partial_t \Phi(\xi, \tau) - P_{\mathrm{HJB}}\Phi(\xi, \tau))^2\right]. \quad (17)$$

## 3 EXPERIMENTS

In the following section, we evaluate the above losses and regularizers on different benchmarks. Specifically, we consider the following losses:

$$\begin{aligned}
&\text{ODE / SDE:} && \mathcal{L}_{\mathrm{FP}}(w, z, \mu) && (18)\\
&\text{ODE-anneal:} && \mathcal{L}_{\mathrm{anneal}}(z, \mu) && (19)\\
&\text{SDE-diffusion:} && \mathcal{L}_{\mathrm{diff}}(w, z) && (20)\\
&\text{OT / SB:} && \mathcal{L}_{\mathrm{FP}}(w, z, \nabla\Phi) + \lambda\mathcal{L}_{\mathrm{HJB}}(\Phi), \quad \lambda > 0. && (21)
\end{aligned}$$

For the benchmarks, we follow Richter et al. (2023) and consider the following Gaussian mixture model and Many-Well distribution.

**Gaussian mixture model (GMM):** We consider the density

$$q_{\mathrm{target}}(x) = p_{\mathrm{target}}(x) = \frac{1}{m}\sum_{i=1}^m \mathcal{N}(x; \mu_i, \Sigma_i). \quad (22)$$

Following Zhang & Chen (2022), we choose $m = 9$, $\Sigma_i = 0.3\,\mathrm{I}$, and

$$(\mu_i)_{i=1}^9 = \{-5, 0, 5\} \times \{-5, 0, 5\} \subset \mathbb{R}^2 \quad (23)$$

to obtain well-separated modes.

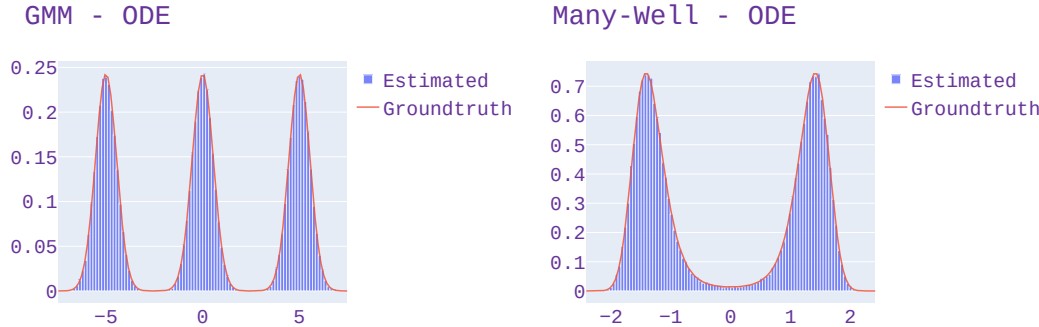

Figure 1: The groundtruth marginal in the first dimension and histograms of samples from our best-performing ODE method on the 2-dimensional GMM (**left**) and 50-dimensional Many-Well (**right**) examples.

**Many-Well (MW):** A typical problem in molecular dynamics considers sampling from the stationary distribution of a Langevin dynamics. In our example we shall consider a $d$-dimensional *many-well* potential, corresponding to the (unnormalized) density

$$q_{\text{target}}(x) = \exp\left(-\sum_{i=1}^{m}(x_i^2 - \delta)^2 - \frac{1}{2}\sum_{i=m+1}^{d} x_i^2\right) \tag{24}$$

with $m \in \mathbb{N}$ combined double wells and a separation parameter $\delta \in (0, \infty)$, see also Wu et al. (2020); Berner et al. (2022). Note that, due to the many-well structure of the potential, the density contains $2^m$ modes. For these multimodal examples, we can compute reference solutions by numerical integration since $q_{\text{target}}$ factorizes in the dimensions.

We compare against the *Path Integral Sampler* (PIS) (Zhang & Chen, 2022) and the *Time-Reversed Diffusion Sampler* (DIS) (Berner et al., 2022), including the log-variance loss by Richter et al. (2023). We refer to Appendix C and Appendix D for further details on our experiments.

Our results are summarized in Table 1. Comparing our losses in (18)–(21), we observe that the ODE method typically works best. Moreover, we can also outperform our baselines on several metrics. In Figure 1 we also see that it accurately covers the modes of the distributions. On the other hand, we observe that the HJB regularization does not provide substantial improvements, and there is no clear advantage of the methods with prescribed density (i.e., SDE-diffusion and ODE-anneal). This indicates that, in general, the non-uniqueness of the ODE/SDE-objectives does not seem to hurt performance. Finally, the additional computational complexity of considering SDE-based (as opposed to ODE-based) methods appears not to pay off.

## 4 CONCLUSION

We provide a framework for using PINNs to sample from unnormalized densities. In particular, we propose to learn the controls of SDEs or ODEs as solutions to (systems of) PDEs that govern their densities. First, this provides a unifying PDE perspective on various sampling methods that are based on normalizing flows, diffusion models, optimal transport, and Schrödinger bridges. Moreover, it yields flexible objectives that are free of time-discretizations and simulations.

We benchmark our methods on multimodal target distributions with up to 50 dimensions. While some SDE-based methods are still unstable, ODE-based variants yield competitive methods that can outperform various baselines. We anticipate that our methods can be improved even further using combinations with simulation-based losses as well as common tricks for PINNs, see Appendix D.

Table 1: Metrics for the Gaussian mixture distribution and the Many-Well in two dimensions $d$. We report errors for estimating the log-normalizing constant $\Delta \log Z$ as well the standard deviations $\Delta \operatorname{std}$ of the marginals. Furthermore, we report the normalized effective sample size ESS and the Sinkhorn distance $\mathcal{W}_\gamma^2$ (Cuturi, 2013), see Appendix D.1 for details. The arrows $\uparrow$ and $\downarrow$ indicate whether we want to maximize or minimize a given metric.

| Problem | Method | $\Delta \log Z \downarrow$ | $\mathcal{W}_\gamma^2 \downarrow$ | ESS $\uparrow$ | $\Delta \operatorname{std} \downarrow$ |
|---|---|---|---|---|---|
| GMM | PIS-KL (Zhang & Chen, 2022) | 1.094 | 0.467 | 0.0051 | 1.937 |
| $(d = 2)$ | PIS-LV (Richter et al., 2023) | 0.046 | **0.020** | 0.9093 | 0.023 |
| | DIS-KL (Berner et al., 2022) | 1.551 | 0.064 | 0.0226 | 2.522 |
| | DIS-LV Richter et al. (2023) | 0.056 | **0.020** | 0.8660 | **0.004** |
| | SDE | 2.140 | 1.507 | **0.9985** | 3.306 |
| | SDE-diffusion | 0.041 | 0.057 | 0.9043 | 0.037 |
| | SB | 3.147 | 0.137 | 0.0005 | 2.717 |
| | ODE | **0.001** | 0.021 | 0.9980 | 0.017 |
| | ODE-anneal | 6.122 | 0.912 | 0.0003 | 3.642 |
| | OT | 0.130 | 0.022 | 0.7874 | 0.497 |
| MW | PIS-KL (Zhang & Chen, 2022) | 3.567 | 1.699 | 0.0004 | 1.409 |
| $(d = 5, m = 5, \delta = 4)$ | PIS-LV (Richter et al., 2023) | 0.214 | 0.121 | 0.6744 | **0.001** |
| | DIS-KL (Berner et al., 2022) | 1.462 | 1.175 | 0.0012 | 0.431 |
| | DIS-LV (Richter et al., 2023) | 0.375 | 0.120 | 0.4519 | **0.001** |
| | SDE | 0.161 | 0.123 | 0.8167 | 0.016 |
| | SDE-diffusion | 3.969 | 0.427 | 0.0124 | 0.004 |
| | SB | 29.095 | 1.565 | 0.0879 | 0.764 |
| | ODE | **0.007** | **0.119** | **0.9904** | 0.007 |
| | ODE-anneal | 0.025 | 0.121 | 0.9506 | 0.005 |
| | OT | 137.66 | 0.403 | 0.0558 | 0.122 |
| MW | PIS-KL (Zhang & Chen, 2022) | 0.101 | 6.821 | 0.8172 | 0.001 |
| $(d = 50, m = 5, \delta = 2)$ | PIS-LV (Richter et al., 2023) | 0.087 | 6.823 | 0.8453 | **0.000** |
| | DIS-KL (Berner et al., 2022) | 1.785 | 6.854 | 0.0225 | 0.009 |
| | DIS-LV (Richter et al., 2023) | 1.783 | 6.855 | 0.0227 | 0.009 |
| | SDE | 0.104 | 6.824 | 0.9027 | 0.003 |
| | SDE-diffusion | 1.989 | **6.803** | 0.1065 | 0.016 |
| | SB | 189.71 | 7.552 | 0.0106 | 0.051 |
| | ODE | **0.038** | 6.820 | **0.9510** | 0.001 |
| | ODE-anneal | 1.759 | 6.821 | 0.2100 | 0.017 |
| | OT | 0.104 | 6.824 | 0.9027 | 0.001 |

ACKNOWLEDGMENTS

J. Sun acknowledges support from Caltech Associates SURF Fellowship. J. Berner acknowledges support from the Wally Baer and Jeri Weiss Postdoctoral Fellowship. A. Anandkumar is supported in part by Bren endowed chair and by the AI2050 senior fellow program at Schmidt Sciences.

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

## A    RELATED WORK

There are numerous Monte Carlo-based methods for sampling from unnormalized densities, including *Markov chain Monte Carlo* (MCMC) (Kass et al., 1998), *Annealed Importance Sampling* (AIS) (Neal, 2001), and *Sequential Monte Carlo* (SMC) (Del Moral et al., 2006; Doucet et al., 2009). However, these methods typically only guarantee *asymptotic* convergence to the target density, with potentially slow convergence rates in practical scenarios (Robert et al., 1999). Variational methods, such as *mean-field approximations* (Wainwright et al., 2008) and *normalizing flows* (Papamakarios et al., 2021), offer an alternative approach. In these methods, the problem of density estimation is transformed into an optimization problem by fitting a parametric family of tractable distributions to the target density. In the context of normalizing flows, we want to mention works on constructing better loss functions (Felardos et al., 2023) or gradient estimators (Vaitl et al., 2022).

In this work, we provide a comprehensive PDE perspective on SDE-based sampling methods. Our approach is loosely inspired by Máté & Fleuret (2023), however, extended to diffusion models, optimal transport (OT), and Schrödinger bridges (SBs). Moreover, we consider other parametrizations and do not rely on the ODE for sampling the collocation points $(\xi, \tau)$. For a corresponding mean-field games (MFG) perspective, we refer to Zhang & Katsoulakis (2023). We also mention path space measure perspectives on SDE-based methods in Vargas & Nüsken (2023); Richter et al. (2023).

The PDE for diffusion models has been derived in Berner et al. (2022) based on prior work by Pavon (1989); Fleming & Rishel (2012) in stochastic optimal control. We refer to Chen et al. (2016) for the corresponding PDEs prominent in OT and SBs. Versions of the Hamilton-Jacobi-Bellman (HJB) regularizer have been used for normalizing flows in generative modeling by Onken et al. (2021), for generalized SBs by Liu et al. (2022); Koshizuka & Sato (2022), for MFG by Ruthotto et al. (2020); Lin et al. (2021), and for generative adversarial models by Yang & Karniadakis (2020).

For the usage of PINNs for a generalized SB in the context of colloidal self-assembly, we refer to Nodozi et al. (2023). An orthogonal direction to our approach is using divergence-free neural networks, which automatically satisfy the continuity equation and only require to fit the boundary distributions $p_\text{target}$ and $p_\text{prior}$ (Richter-Powell et al., 2022). We also mention that higher-dimensional Fokker-Planck equations have also been tackled with time-varying Gaussian mixtures (Chen & Majda, 2018), and there exist SDE-based neural solvers for HJB equations (Richter & Berner, 2022; Nüsken & Richter, 2021b) and combinations with PINNs (Nüsken & Richter, 2021a).

Finally, we want to highlight recent works on simulation-free learning of (stochastic) dynamics using *flow matching* (Tong et al., 2023; Lipman et al., 2022) and *action matching* techniques (Neklyudov et al., 2022). However, these methods rely on samples from the target distribution $p_\text{target}$. Similarly, many works on solving SB and OT problems using deep learning require samples from the target distribution (Chen et al., 2021; De Bortoli et al., 2021; Fernandes et al., 2021; Vargas et al., 2021).

## B    HJB EQUATION

Let us present a sketch of the proof that the optimal drift can be represented as a gradient field, see also Neklyudov et al. (2022); Koshizuka & Sato (2022); Benamou & Brenier (2000); Caluya & Halder (2021). Let us consider the optimization problem

$$\inf_\mu \; \frac{1}{2} \int_0^T \int_{\mathbb{R}^d} \|\mu\|^2 \, p \, \mathrm{d}x \, \mathrm{d}s \tag{25}$$

$$\text{s.t.} \quad \partial_t p = -\operatorname{div}(p\mu) + \frac{1}{2} \operatorname{Tr}(\sigma\sigma^\top \partial_{xx} p), \quad p(\cdot, 0) = p_\text{prior} \quad p(\cdot, T) = p_\text{target}. \tag{26}$$

for a sufficiently smooth density $p$. Introducing a Lagrange multiplier $\Phi \colon \mathbb{R}^d \times [0, T] \to \mathbb{R}$, we can rewrite the problem as

$$\sup_\Phi \inf_\mu \int_0^T \int_{\mathbb{R}^d} \frac{1}{2} \|\mu\|^2 \, p + \Phi \left( \partial_t p + \operatorname{div}(p\mu) - \frac{1}{2} \operatorname{Tr}(\sigma\sigma^\top \partial_{xx} p) \right) \, \mathrm{d}x \, \mathrm{d}s. \tag{27}$$

Using integration by parts, we can calculate

$$\int_0^T \Phi \partial_t p \, \mathrm{d}s = \left[ \Phi p \right]_{s=0}^{s=T} - \int_0^T p \, \partial_t \Phi \, \mathrm{d}s. \tag{28}$$

and

$$\int_{\mathbb{R}^d} \Phi \operatorname{Tr}(\sigma\sigma^\top \partial_{xx} p)\,\mathrm{d}x = \int_{\mathbb{R}^d} p \operatorname{Tr}(\sigma\sigma^\top \partial_{xx}\Phi)\,\mathrm{d}x, \tag{29}$$

where we assume that $p$ and its partial derivatives vanish sufficiently fast at infinity. Using the product rule and Stokes' theorem, we obtain that

$$\int_{\mathbb{R}^d} \Phi \operatorname{div}(p\mu)\,\mathrm{d}x = \int_{\mathbb{R}^d} \operatorname{div}(\Phi p\mu)\,\mathrm{d}x - \int_{\mathbb{R}^d} p\,\mu\cdot\nabla\Phi\,\mathrm{d}x = -\int_{\mathbb{R}^d} p\,\mu\cdot\nabla\Phi\,\mathrm{d}x. \tag{30}$$

Leveraging Fubini's theorem and combining the last three calculations with (27), we obtain that

$$\sup_{\Phi}\inf_{\mu} \int_{\mathbb{R}^d}\int_0^T \left(\frac{1}{2}\|\mu\|^2 - \mu\cdot\nabla\Phi\right)p - \left(\partial_t\Phi + \frac{1}{2}\operatorname{Tr}(\sigma\sigma^\top\partial_{xx}\Phi)\right)p\,\mathrm{d}s + \left[\Phi p\right]_{s=0}^{s=T}\,\mathrm{d}x. \tag{31}$$

In view of the binomial formula, we observe that the minimizer is given by

$$\mu = \nabla\Phi. \tag{32}$$

We can thus write (31) as

$$\inf_{\Phi} \int_{\mathbb{R}^d}\int_0^T \left(\partial_t\Phi + \frac{1}{2}\|\nabla\Phi\|^2 + \frac{1}{2}\operatorname{Tr}(\sigma\sigma^\top\partial_{xx}\Phi)\right)p\,\mathrm{d}s - \left[\Phi p\right]_{s=0}^{s=T}\,\mathrm{d}x, \tag{33}$$

which corresponds to the action matching objective in Neklyudov et al. (2022). We also refer to Neklyudov et al. (2022) for existence and uniqueness results. If we additionally minimize (25) over all densities $p$ with $p(\cdot,0) = p_{\mathrm{prior}}$ and $p(\cdot,T) = p_{\mathrm{target}}$, we obtain the problem

$$\inf_{\Phi,p} \int_{\mathbb{R}^d}\int_0^T \left(\partial_t\Phi + \frac{1}{2}\|\nabla\Phi\|^2 + \frac{1}{2}\operatorname{Tr}(\sigma\sigma^\top\partial_{xx}\Phi)\right)p\,\mathrm{d}s - \left[\Phi p\right]_{s=0}^{s=T}\,\mathrm{d}x, \tag{34}$$

Computing the functional derivative w.r.t. $p$, we obtain the first-order optimality condition

$$\partial_t\Phi = -\frac{1}{2}\operatorname{Tr}(\sigma\sigma^\top\partial_{xx}\Phi) - \frac{1}{2}\|\nabla\Phi\|^2, \tag{35}$$

which yields the HJB equation in (16).

## C  LOG-LIKELIHOODS AND IMPORTANCE WEIGHTS

This section describes ways to compute the log-likelihood and importance weights for samples $X_T$ obtained from the stochastic process $X$.

**ODEs:**  In the setting of normalizing flows, we can compute the evolution of the log-density along the trajectories. Using $\frac{\mathrm{d}}{\mathrm{d}t}X_t = \mu(X_t,t)$, we can show that

$$\frac{\mathrm{d}}{\mathrm{d}t}v(X_t,t) = (\nabla v\cdot\mu - \operatorname{div}(\mu) - \nabla v\cdot\mu)(X_t,t) = -\operatorname{div}(\mu)(X_t,t), \tag{36}$$

which is often referred to as the *change-of-variables formula*. Recalling that $v = \log p_X$, we can then compute the (unnormalized) *importance weights*

$$w^{(k)} := \frac{q_{\mathrm{target}}}{p_{X_T}}(X_T^{(k)}) \tag{37}$$

of samples $(X_T^{(k)})_{k=1}^K$.

**SDEs:**  If we have (an approximation to) the score $\nabla v = \nabla\log p_X$ of an SDE $X$, we can transform it into an ODE with the same marginals using

$$\mu_{\mathrm{ODE}} = \mu_{\mathrm{SDE}} - \frac{1}{2}\sigma\sigma^\top\nabla v. \tag{38}$$

The above relation can be verified via the Fokker-Planck equation (4), and the resulting ODE is often referred to as *probability flow* ODE (Song et al., 2020). Note that this also allows us to use the change-of-variables formula in (36) for SDEs.

The log-likelihoods can be simulated together with the ODE in (3) and allow us to compute importance weights in the target space $\mathbb{R}^d$. If the optimal drift of the SDE can be described via a change of path measures, such as for the annealed flows (Vargas & Nüsken, 2023) or diffusion models (Berner et al., 2022), we can also perform importance sampling in path space $C([0,T],\mathbb{R}^d)$.

# D EXPERIMENTS

In this section, we describe our metrics as well as our implementation.

## D.1 METRICS

We evaluate the performance of our methods on the following metrics.

**Normalizing constants:** We could obtain an estimate $\log z(T)$ of the log-normalizing constant $\log Z$ by our parametrization in (6). However, since we are interested in the sample quality of our models, we use the log-likelihood to compute a lower bound for $\log Z$, see Appendix C. Note that we do not employ importance sampling for estimating the log-normalizing constant.

**Standard deviations:** We also analyze the error when approximating coordinate-wise standard deviations of the target distribution $p_{\text{target}}$, i.e.,

$$\frac{1}{d} \sum_{k=1}^{d} \sqrt{\mathbb{V}[X_i]}, \quad \text{where} \quad X \sim p_{\text{target}}, \tag{39}$$

using samples $(X_T^{(k)})_{k=1}^{K}$ from our model.

**Effective sample size:** One would like to have the variance of the importance weights small, or, equivalently, maximize the (normalized) *effective sample size*

$$\text{ESS} := \frac{\left( \sum_{k=1}^{K} w^{(k)} \right)^2}{n \sum_{k=1}^{K} (w^{(k)})^2}. \tag{40}$$

The computation of the importance weights is outlined in Appendix C.

## D.2 IMPLEMENTATION

**Networks:** We use a Fourier-MLP as in Zhang & Chen (2022) for the networks $w$. For the network $\mu$, we experimented with both Fourier-MLPs and standard MLPs with residual connections. For ODE-anneal we also parametrize $z$ by a small Fourier-MLP. For the other methods, $z$ does not need to depend on $t$, and we just use a single trainable parameter.

**Parameters:** We choose $T = 1$ and $p_{\text{prior}} = \mathcal{N}(0, \text{I})$. We set $\sigma$ to a constant value, i.e., $\sigma(x, t) = \bar{\sigma} \text{I}$, where $\bar{\sigma} \in \{0, \sqrt{2}\}$. For the diffusion model, we pick a simple VP-SDE from Song et al. (2020) with $f(x, t) := -\frac{\bar{\sigma}^2}{2} x$ to satisfy that $p_Y(\cdot, T) \approx p_{\text{prior}}$ (for sufficiently large $\bar{\sigma}$ and $T$).

**Training and inference:** We train with batch-size 4096 for $200k$ gradient steps using the Adam optimizer with exponentially decaying learning rate. For simulating the SDEs and ODEs during inference, we use the Euler-Maruyama and Fourth-order Runge-Kutta (with 3/8 rule) scheme, respectively. We use $100k$ samples to evaluate our methods. Finally, we performed a grid-search over the penalty parameter $\lambda$ of the HJB loss $\mathcal{L}_{\text{HJB}}$ and over the initial learning rate as well as its decay per step.

# E LIMITATIONS AND EXTENSIONS

In this section, we mention potential limitations and extensions of our framework.

## E.1 SAMPLING

Let us investigate two choices of how to choose the random variables $(\xi, \tau)$ to penalized the loss in (7). We will show how these choices allow to balance exploration and exploitation.

**Uniform** We can simply chose $\tau \sim \mathrm{Unif}([0, T])$ and $\xi \sim \mathrm{Unif}(K)$ for a sufficiently large compact set $K \subset \mathbb{R}^d$. This choice allows us to uniformly explore the domain $K$, which is particularly interesting at the beginning of the training. Moreover, different from most other methods, we do not need to rely on (iterative) simulations of the SDE in (3). In order to specify $K$, however, we need prior information to estimate the domain where $v$ is above some minimal threshold. We incur an approximation error if the set $K$ is chosen too small. On the other hand, if it is too large, low probability areas of $p_{\mathrm{target}}$ can lead to instabilities and might require clipping. This could be circumvented by picking a distribution $\xi$ that is supported on the whole spatial domain $\mathbb{R}^d$. Alternatively, we can use adaptive methods as outlined in the next paragraphs.

**Along the Trajectories** We can also simulate the SDE using the partially learned drift coefficient $\mu$ to exploit the learned dynamics. This corresponds to the choices $\tau \sim \mathrm{Unif}([0, T])$ and $\xi \sim X_\tau$. Note that we just use the SDE/ODE for sampling the collocation points, and we are not backpropagating through the solver (to update the drift $\mu$). In other words, we detach $\xi$ from the computational graph.

Moreover, we want to mention improved sampling strategies for PINNs, see, e.g., Tang et al. (2023); Chen et al. (2023). Similar to Quasi-Monte Carlo methods, one could also leverage low-discrepancy samplers for the time coordinate $\tau$, as, e.g., used by Kingma et al. (2021).

### E.2    PINNs

It is commonly known that PINNs can be sensitive to hyperparameter settings. We can make use of a plethora of tricks that have been proposed to stabilize their training (Wang et al., 2023). For instance, for the neural networks, one could additionally consider random weight factorization and Fourier features for the spatial coordinates. Moreover, we can choose the penalty parameter $\lambda$ for the HJB loss $\mathcal{L}_{\mathrm{HJB}}$ adaptively based on the residuals and their gradients. We also mention that the computation of divergences and Laplacians using automatic differentiation can be prohibitive in very high dimensions and might require (stochastic) estimators, such as Hutchinson's trace estimator. Alternatvely, we could also explore the OT-Flow architecture for $\Phi$, which has been successfully employed by Onken et al. (2021); Koshizuka & Sato (2022); Ruthotto et al. (2020).

### E.3    Noise Schedule

We can consider time-dependent diffusion coefficients $\sigma$, which have been successfully employed for diffusion models. For instance, we can adapt the VP-SDE in Song et al. (2020) with

$$\bar{\sigma}(t) \coloneqq \sqrt{2\beta(t)}\,\mathrm{I} \quad \text{and} \quad \overleftarrow{f}(x, t) \coloneqq -\beta(t)x, \tag{41}$$

where

$$\beta(t) \coloneqq \frac{1}{2}\left(\left(1 - \frac{t}{T}\right)\sigma_{\min} + \frac{t}{T}\sigma_{\max}\right). \tag{42}$$

Our framework also allows for diffusion coefficients $\sigma$ which depend on the spatial coordinate $x$. Finally, we could also learn the diffusion, for instance, using the parametrization $\sigma = \mathrm{diag}(\exp(s))$ for a neural network $s$.

### E.4    Mean-Field Games

More generally, we could extend our framework to (stochastic) mean-field games (MFG), mean-field control problems, and generalized SBs (Benamou et al., 2017; Zhang & Katsoulakis, 2023; Liu et al., 2022; Lin et al., 2021; Koshizuka & Sato, 2022; Ruthotto et al., 2020).

