# OpenReview forum: "Physics-informed neural networks for sampling"
_ICLR.cc/2024/Workshop/AI4DiffEqtnsInSci — AI4DiffEqtnsInSci @ ICLR 2024 Poster_

### Official Review · Reviewer_r3JV · 2024-02-20

**Rating:** 7
**Confidence:** 2

**Review:**

This paper lays out a framework for ODE and SDE distributional sampling using PINNs. It presents the loss terms for different evolution models, and compares PINN to path integral samplers. This is an interesting idea, it's great to see PINNs being used in domains outside of physical modeling. This isn't quite my area, so I can't speak to whether there's similar existing work though.
The benchmarks are rather low-dimensional and simple, but that's fine for an exploration in the context of a workshop.

---

### Meta-Review · Program_Chairs · 2024-03-03

**Recommendation:** Accept (Poster)

**Metareview:**

The reviewer marks this paper as a clear accept and I agree. It is a nice and novel application of PINNs.

---

### Decision · Program_Chairs · 2024-03-03

Accept (Poster)